# DecETT: Accurate App Fingerprinting Under Encrypted Tunnels via Dual Decouple-based Semantic Enhancement

## Abstract

Due to the growing demand for privacy protection, encrypted tunnels have become increasingly popular among mobile app users, which brings new challenges for app fingerprinting (AF)-based network management. Existing methods primarily transfer traditional AF methods to encrypted tunnels directly, ignoring the core obfuscation and re-encapsulation mechanism of encrypted tunnels, thus resulting in unsatisfactory performance. In this paper, we propose DecETT, a dual decouple-based semantic enhancement method for accurate AF under encrypted tunnels. Specifically, DecETT improves AF under encrypted tunnels from two perspectives: app-specific feature enhancement and irrelevant tunnel feature decoupling. Considering the obfuscated app-specific information in encrypted tunnel traffic, DecETT introduces TLS traffic with stronger app-specific information as a semantic anchor to guide and enhance the fingerprint generation for tunnel traffic. Furthermore, to address the app-irrelevant tunnel feature introduced by the re-encapsulation mechanism, DecETT is designed with a dual decouple-based fingerprint enhancement module, which decouples the tunnel feature and app semantic feature from tunnel traffic separately, thereby minimizing the impact of tunnel features on accurate app fingerprint extraction. Evaluation under five prevalent encrypted tunnels indicates that DecETT outperforms state-of-the-art methods in accurate AF under encrypted tunnels, and further demonstrates its superiority under tunnels with more complicated obfuscation. *Project page: https://github.com/DecETT/DecETT*

## Keywords

App Fingerprinting, Encrypted Tunnel, Encrypted Traffic Analysis, Decouple-based Representation Learning

## 1 Introduction

Over the past few years, we have witnessed the widespread use of encrypted tunnels in mobile network communications[10, 34, 41]. Serving as intermediaries that forward traffic between apps and servers, encrypted tunnels conceal both the identities of the communicating parties and the transmitted traffic characteristics, thus providing an effective way for privacy protection[3] and anonymous communication[25]. However, the prevalence of encrypted tunnels also poses new challenges to network management, such as Quality of Service (QoS)[42] and behavior auditing[1]. Traditional network management strategies primarily rely on app fingerprinting (AF) that identifies app usage activities by analyzing server information[26, 33] (e.g., IP address or Server Name Indicator) or TLS traffic characteristics[20, 30, 37]. However, encrypted tunnels obfuscate these two distinctive features, making accurate AF more challenging than in the traditional TLS scenario.

While prior work has developed some AF methods under encrypted tunnels, most of them directly transfer traditional AF methods, ignoring the core impact caused by tunnel mechanism. As

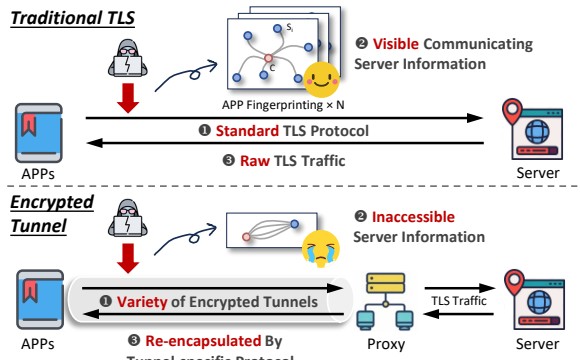

**Figure 1: Three main challenges in App Fingerprinting under encrypted tunnels: (1) Diversity of encrypted tunnels, (2) Server concealment, and (3) Traffic re-encapsulation.**

illustrated in Figure 1, there exist three primary challenges in employing AF under encrypted tunnels compared with traditional TLS scenarios. **(1) Diversity of encrypted tunnels.** Currently, there are numerous kinds of encrypted tunnels that have been widely used. Some studies design effective AF methods for specific encrypted tunnels[9, 14, 35], such as Shadowsocks and SSH. However, since different encrypted tunnels employ varying forwarding policies and encapsulation protocols for the original TLS traffic, developing specific AF methods for each tunnel type is labor-intensive and inefficient. **(2) Lack of server information.** In traditional TLS scenarios, server information, such as IP addresses, TLS certificates, and high-level interaction patterns, can be directly extracted from the original TLS traces to facilitate fingerprint construction. However, in encrypted tunnels, all traffic is forwarded to the tunnel server instead of the actual app servers, concealing any server-related information. As a result, server information-based methods, which perform excellently in TLS scenario[5, 26, 33], cannot be applied under encrypted tunnels. **(3) Weaker AF semantic representations caused by re-encapsulation.** Existing methods attempt to extract discriminative features directly from the tunnel traffic[17, 21, 24, 39, 43]. However, encrypted tunnels employ re-encapsulation mechanism on the forwarded TLS traffic to ensure the confidentiality of tunnel communication. This process not only obfuscates the raw app-specific information, but also introduces tunnel-related information that are irrelevant to apps into the tunnel traffic, resulting in unsatisfactory performance and making accurate AF more challenging.

To address the aforementioned issues, in this paper, we propose DecETT, a dual decouple-based semantic enhancement method for accurate AF under encrypted tunnels. DecETT utilizes flow sequences as the representation form of traffic to avoid the limitations of inaccessible server information. Specifically, DecETT consists

of three key steps as follows. Firstly, to mitigate the obfuscated app-specific features caused by re-encapsulation, we introduce TLS traffic as a stronger and more stable semantic anchor to guide and enhance the fingerprint generation for tunnel traffic. Each tunnel flow is correlated with its corresponding original TLS flow for further analysis. Secondly, to address the negative impact of tunnel-related features within tunnel traffic, DecETT incorporates a dual decouple-based fingerprint enhancement module, which adopts a dual-branch Siamese network to tackle TLS and tunnel traffic separately. By decoupling the disentangled protocol features and app semantic features within the traffic, DecETT isolates protocol-related features that are irrelevant to app fingerprints, therefore reducing the impact of the re-encapsulation mechanism on capturing distinguishable app-specific information. Finally, the app semantic features extracted from tunnel traffic are input into the classifier as the generated fingerprint for the final AF results. To validate the effectiveness of DecETT, we conduct extensive experiments under five widely used encrypted tunnels.

**Contributions.** Our contributions can be summarized as:

- We propose a dual decouple-based semantic enhancement method, DecETT, which can achieve accurate app fingerprinting under various encrypted tunnels.
- Considering the obfuscation of app-specific information caused by re-encapsulation, we introduce TLS traffic with stronger and more stable app semantic information to guide and enhance effective fingerprint generation.
- We design a dual decouple-based semantic enhancement module to decouple tunnel-related features and app-specific semantic features, which mitigates the negative impact of re-encapsulation on accurate fingerprint extraction.
- Evaluated under five widely-used encrypted tunnels, DecETT outperforms state-of-the-art methods on multiple metrics, and shows superiority under tunnels with more complicated obfuscation.

The remainder of this paper is organized as follows. Section 2 summarizes the prior research related to our work. Section 3 introduces the necessary foundational knowledge of this paper. Section 4 highlights the overall design of DecETT, and Section 5 illustrates the experiments. Section 6 concludes the paper.

## 2 Related Work

From the task perspective, prior relevant works mainly focus on app fingerprinting and encrypted tunnel traffic analysis, respectively. In this section, we briefly review and discuss these works.

### 2.1 App Fingerprinting

App Fingerprinting (AF) refers to a side-channel network management technique that identifies app usage activities through encrypted traffic analysis. Although the packet payloads are encrypted, certain traffic characteristics, such as server profiles, TLS certificates, and flow sequences, still allow for successful AF under encrypted traffic. Generally, prior works can be categorized into two main groups, including server information analysis and flow feature mining.

**Server Information Analysis.** Server information analysis-based methods refer to using server-related features for accurate AF.

Van Ede et al. [33] and Pham et al. [26] explore temporal correlations among destination-related features of network traffic and use these correlations to generate app fingerprints.

**Flow Feature Mining.** These methods focus on extracting fingerprints from the transmitted traffic flows and can be further divided into three parts. A typical approach is to utilize statistical features that are independent of encryption, such as packet lengths[30] and time-related features[4]. Another kind of approach[36, 38, 43] extracts the raw bytes of packets and employs deep learning to identify distinguishable app features based on the pseudo-randomness of encryption algorithms. For instance, ET-Bert[17] transforms the packet payloads into word-like tokens and achieves satisfactory performance based on the pre-training technique. Besides, deep mining of flow sequences also provides effective AF strategies. Liu et al. [18] utilize multi-layer end-to-end encoder-decoder structure to mine the potential sequence characteristics. Shen et al. [28] construct each flow sequence as a graph by burst division and association, and transform AF to a graph classification task.

While these methods have demonstrated high accuracy in traditional TLS scenario, their performance diminishes under encrypted tunnels since both server information and traffic characteristics are obfuscated, making effective AF more challenging.

### 2.2 Encrypted Tunnel Traffic Analysis

Currently, works for encrypted tunnel traffic analysis mainly focus on detecting tunnel flows from a massive amount of traffic. Several studies [16, 19, 22, 23] analyze and extract tunnel-specific protocol features to achieve accurate identification. For instance, Xue et al. [41] constructs OpenVPN traffic fingerprints from the aspects of byte pattern, packet size, and server response to achieve accurate OpenVPN traffic identification.Alice et al. [2] observe that the length and entropy value of the first packet in a flow can be used as specific features for Shadowsocks traffic detection, and combine active probing to further improve the identification accuracy.

Some other methods [8, 13, 24] dive into AF under encrypted tunnels for more fine-grained analysis. Xu et al. [40] convert each tunnel flow into a graph and combine it with statistical features to realize app classification. Wang et al. [35] add the sliding window JS divergence feature based on the traditional packet length and timestamp-related statistics to promote the accuracy and robustness of AF under Shadowsocks.

In summary, existing AF methods under encrypted tunnels mainly follow the technical roadmap of traditional TLS traffic classification and lack targeted solutions for the tunnel mechanism. Therefore, their performance is still limited by the weak app semantic features in tunnel traffic. In this work, we aim to mitigate the negative impact of tunnel obfuscation by both irrelevant tunnel feature decoupling and app semantic feature enhancement with the help of TLS traffic, thereby achieving accurate AF under various tunnels.

## 3 Preliminaries

In this section, we first provide the threat model of app fingerprinting, and then conduct a detailed analysis of the core principle of the tunnel re-encapsulation mechanism and its impact on tunnel flow sequences, to provide the necessary theoretical foundation.

```
1   # socket init and connection establishment
2   local_sock = TCPRelay.socket.accept()
3   remote_sock = create_remote_socket(server_ip, server_port)
4
5   # read raw data from local
6   data = local_sock.recv(BUF_SIZE)
7
8   # encrypte and send data
9   data = encryptor.encrypt(data)
10  write_to_sock(data, remote_sock)
11
12  # receive and decrypt data from remote
13  data = remote_sock.recv(BUF_SIZE)
14  data = encryptor.decrypt(data)
15
16  # send raw data to local
17  write_to_sock(data, local_sock)
```

**Figure 2: Source code of re-encapsulation mechanism summarized from Shadowsocks. Illustrations for the other 4 encrypted tunnels can be found in Appendix A.**

### 3.1 Threat Model

In this paper, we refer to the threat model [26, 33] in previous app fingerprinting studies, with the critical difference that we focus on the more complex encrypted tunnel scenario. Specifically, an app fingerprinting system is located at the network boundary, where it can collect and analyze all traffic sent out from this network. The primary goal of AF is to identify app usage activities concealed in encrypted tunnels of specific mobile devices by analyzing the corresponding tunnel traffic. We assume that only one app is executed at a time, i.e., composite app fingerprints are not considered[33].

### 3.2 Re-encapsulation Mechanism

Firstly, to reveal the principle of the re-encapsulation mechanism intuitively, we summarize the source code of traffic re-encapsulation and forwarding process in Shadowsocks[27], a widely used encrypted tunnel tool for mobile devices, as shown in Figure 2. Unnecessary functions and parameters are omitted, and annotations are added for clarity. When a local app attempts to send data through the tunnel, the tunnel client establishes a connection with the local app via *local_sock* and creates a corresponding *remote_sock* to the tunnel server simultaneously. The tunnel client then receives and encrypts the data from the local app according to the tunnel protocol, and forward it to the tunnel server. Similarly, upon receiving responses from the tunnel server, the tunnel client decrypts the data and sends it back to the local app, thereby achieving traffic forwarding of encrypted tunnels.

In summary, the tunnel client maintains two TCP connections and their correlation: one for communicating with the local app called *inbound* and another for data transmission with the tunnel server called *outbound*. Both connections are implemented via socket communication, so the process of data encryption and forwarding does not vary based on the class of app data. Therefore, tunnel protocol features and app semantic features can be viewed as two independent variables for tunnel traffic generation, thereby ensuring the feasibility of the feature decoupling.

### 3.3 Impact on Tunnel Flow Sequences

Based on the principle of the re-encapsulation mechanism above, this section discusses its impact on tunnel flow sequences. We select a TLS flow and its two corresponding tunnel flows forwarded by V2Ray[31] for comparison.

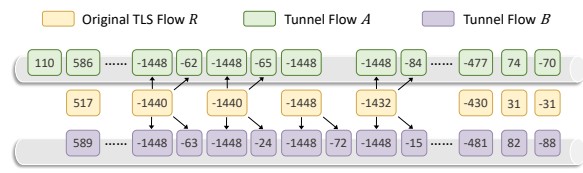

**Figure 3: Flow sequence variation caused by tunnel re-encapsulation mechanism.**

As shown in Figure 3, the tunnel flow sequences differ from the TLS flow sequence after being forwarded by the tunnel. This variation can be attributed to the fact that tunnel flow sequences are affected by both the original app and tunnel re-encapsulation. Specifically, the impact of the latter on flow sequences mainly lies in three aspects. **(1) Packet length variation.** Compared to the original TLS traffic, the packet lengths in both flow $A$ and $B$ increase to varying degrees due to the additional byte overhead caused by tunnel re-encapsulation. Furthermore, the same TLS packet may correspond to packets of different lengths after re-encapsulation. For example, a packet with a length of 517 in flow $R$ corresponds to packets of 586 and 589 in flows $A$ and $B$, respectively. **(2) Packet fragmentation.** Due to the extra byte overhead and the limitation of the Maximum Transmission Unit (MTU), the payload data of a single TLS packet may be split into two packets for transmission in tunnel traffic. For instance, a packet with a payload of 1440 bytes in flow $R$ is split into two packets of 1448 bytes and 62 bytes in flow $A$. **(3) Packet redundancy.** Some packets, such as the first packet in flow $A$ with a payload of 110 bytes, are not generated by the upper-layer app and are more likely to serve as control packets for tunnel communication.

These variations indicate that tunnel mechanism obfuscate the app-specific information hidden in the flow sequences, resulting in poor AF performance. Considering that TLS traffic remains unaffected by the tunnel mechanism and shares the same app-specific information with tunnel traffic, it can serve as a robust semantic anchor for learning representative app semantic features in tunnel traffic, thereby facilitating accurate AF under encrypted tunnels.

## 4 Design of DecETT

Based on the aforementioned analysis of tunnel mechanism, in this section, we introduce our dual decouple-based semantic enhancement app fingerprinting method, DecETT. As shown in Figure 4, the architecture of DecETT could be divided into three main processes: traffic preprocess and correlation, dual decouple-based fingerprint enhancement, and generated AF classification.

### 4.1 Traffic Preprocess and Correlation

DecETT utilizes TLS traffic as a semantic anchor to mitigate app semantics loss and enhance the representation learning of the tunnel traffic. In this process, we construct parallel correlation flow pairs from the obfuscated network traffic to facilitate subsequent work.

Firstly, we reassemble TLS and tunnel flows separately based on 5-tuple information of the packets, including source IP, source port ($S_{Port}$), destination IP, destination port ($D_{Port}$), and protocol, and then pad or truncate them to the unified flow sequence length $n$.

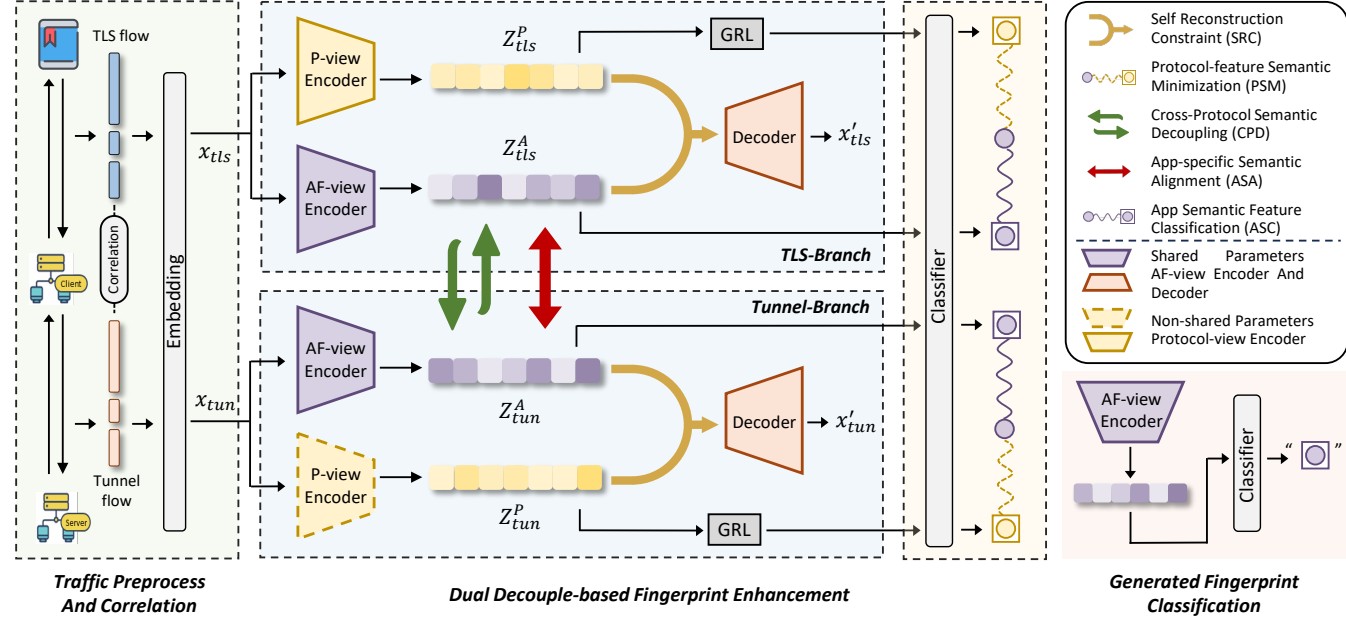

**Figure 4: The overall architecture of DecETT.**

Next, the reassembled TLS and tunnel traffic flows are correlated according to the mapping table $T$ maintained by the tunnel client. As we mentioned in Section 3.2, the tunnel client maintains a socket mapping relation $(inbound, outbound) \in T$ for each pair of the forwarded traffic. The *inbound* keyword records the $S_{Port}$ of the TCP connection established with the app, while the *outbound* keyword records the $D_{Port}$ of the TCP connection established with the tunnel server. Therefore, TLS flow that satisfies $S_{Port} == inbound$ and tunnel flow that satisfies $S_{Port} == outbound$ share the same app-specific information and are correlated as a parallel flow pair. Moreover, in order to avoid the confusion caused by port reuse, we restrict the time difference between the two flow start timestamps $t_{F_{tls}}$, $t_{F_{tun}}$ of the correlated flows to be less than a certain threshold $\varepsilon$. In summary, the flow correlation process can be formally described as the concatenation of $F_{tls}$ and $F_{tun}$ that satisfy:

$$\begin{cases} (S_{port}^{tls}, S_{port}^{tun}) == (inbound, outbound) \in M \\ |t_{F_{tls}} - t_{F_{tun}}| \leq \varepsilon \end{cases} \quad (1)$$

By correlating each tunnel flow with its corresponding TLS flow that shares the same app-specific information, an additional semantic supervisory signal is provided for fingerprint learning of the tunnel flow, thereby facilitating accurate AF under encrypted tunnels.

Furthermore, in order to enrich the information retained in the packets, each parallel flow pair $F_{tls-tun}$ is mapped through a trainable embedding layer $Emb(\cdot)$. Formally, given a flow pair sequence as $F_{tls-tun} = \{[p_{1,tls}, \ldots, p_{n,tls}], [p_{1,tun}, \ldots, p_{n,tun}]\}$, the embedding layer $Emb(\cdot)$ maps each packet $p_i$ to an embedding vector $e_i$ of dimension $d$. Therefore, the raw flow pair is mapped to representation $x_{tls-tun} = [[e_{1,tls}, \ldots, e_{n,tls}], [e_{1,tun}, \ldots, e_{n,tun}]] \in \mathbb{R}^{2n \times d}$ for further analysis.

## 4.2 Dual decouple-based Fingerprint Enhancement

As we discussed in Section 3.2, the representation of encrypted tunnel traffic is jointly influenced by both app semantic and tunnel protocol features. Therefore, irrelevant protocol features inevitably hinder the learning of accurate app semantic features and thus bring negative impact to AF. In this process, we aim to decouple the protocol and app semantic features entangled in the traffic, and further enhance the semantic features with the help of TLS traffic to reduce the negative impact of tunnel re-encapsulation.

Specifically, DecETT employs a partially parameter-shared Siamese Network[7] with two branches to process TLS traffic and tunnel traffic separately. Each branch comprises a protocol-view encoder $Enc^P$, an AF-view encoder $Enc^A$, a decoder $Dec$, and a classifier $C$. Each of the encoders and decoders utilizes a 2-layer stacked Bi-GRU[6] as the backbone to model the contextual bidirectional information of the flow sequences. The protocol-view encoder aims at learning protocol features $Z^P = Enc^P(x)$ that are independent of the app, while the AF-view encoder focuses on extracting app semantic features $Z^A = Enc^A(x)$ from the raw traffic. In order to facilitate the decoupling between these two representations and enhance accurate fingerprint extraction, we propose two specific sub-modules to train DecETT. In the following, we present each of them in detail.

*4.2.1 Flow Representation Decoupling.* In this sub-module, we force the app semantic features to be decoupled with the tunnel protocol features to minimize the negative influence caused by tunnel re-encapsulation.

**Self Reconstruction Constraint (SRC).** Since the original flow representation is decoupled into two independent features $Z^P$ and $Z^A$, it is essential to ensure that these two features retain as much of

the original flow information as possible. Therefore, we first introduce self-reconstruction loss to ensure the fundamental correctness of feature decoupling, which can be calculated as follows:

$$x'_{i,tls} = Dec(Z^P_{i,tls}, Z^A_{i,tls}) \tag{2}$$

$$x'_{i,tun} = Dec(Z^P_{i,tun}, Z^A_{i,tun}) \tag{3}$$

$$\mathcal{L}_{SRC} = -\frac{1}{N}\sum_{i=1}^{N}(||x'_{i,tls} - x_{i,tls}||^2 + ||x'_{i,tun} - x_{i,tun}||^2) \tag{4}$$

where $N$ stands for the total number of flow pairs. By minimizing the difference between the reconstructed and the original flow representations, SR loss constrains the two decoupled features to fully preserve essential characteristics of the original traffic flow, ensuring that no critical information in the original flow is lost during feature decoupling.

**Protocol-feature Semantic Minimization (PSM).** Based on the assurance from SRC that app-specific information is preserved by either $Z^A$ or $Z^P$, minimizing the app-specific information in $Z^P$ equals to maximizing the app-specific information captured by $Z_A$. To achieve this goal, we propose the Protocol-feature Semantic Minimization cross-entropy loss for $Z^P$ under the fingerprint classification task. Suppose $\hat{y}^P_i$ is the predicted app label of $Z^P_i$, the PSM loss can be formulated as

$$\mathcal{L}_{PSM} = -\frac{1}{N}\sum_{i=1}^{N} y_i(log(\hat{y}^P_{i,tls}) + log(\hat{y}^P_{i,tun})) \tag{5}$$

During the training process, we apply Gradient Reversal Layer (GRL) [11] to reverse the gradient during back-propagation to maximize $\mathcal{L}_{PSM}$, thereby minimizing the app-specific information captured by $Z^P$. Under the dual constraints of both SRC and PSM, DecETT encourages $Z^A$ to capture more app-specific information, thereby achieving effective feature decoupling and extraction.

**Cross-Protocol Semantic Decoupling (CPD).** To further facilitate the feature decoupling, we propose the cross-protocol semantics decoupling that swaps the extracted app semantic features $Z^A$ to reconstruct the original flow representations $x_{tls}$ and $x_{tun}$ together with $Z^P_{tls}$ and $Z^P_{tun}$, respectively. Formally, the cross-protocol reconstruction process can be described as follows:

$$\hat{x}_{i,tls} = Dec(Z^P_{i,tls}, Z^A_{i,tun}) \tag{6}$$

$$\hat{x}_{i,tun} = Dec(Z^P_{i,tun}, Z^A_{i,tls}) \tag{7}$$

Thus the CPD loss can be calculated as:

$$\mathcal{L}_{CPD} = -\frac{1}{N}\sum_{i=1}^{N}(||\hat{x}_{i,tls} - x_{i,tls}||^2 + ||\hat{x}_{i,tun} - x_{i,tun}||^2) \tag{8}$$

By minimizing CPD loss, DecETT not only reduces the amount of protocol information irrelevant to apps contained in $Z^A$, but also implicitly aligns the app semantic features extracted from the parallel flow pairs.

Therefore, the total loss of flow representation decoupling submodule $\mathcal{L}_{FRD}$ can be summarized as:

$$\mathcal{L}_{FRD} = \lambda_1\mathcal{L}_{SRC} + \lambda_2\mathcal{L}_{PSM} + \lambda_3\mathcal{L}_{CPD} \tag{9}$$

#### 4.2.2 App Semantic Feature Augmentation.
In this sub-module, the app semantic features decoupled from the tunnel traffic are further augmented by aligning with two supervisory signals with strong semantics.

**App-specific Semantic Alignment (ASA).**

Based on the two decoupled features, ASA explicitly aligns the app semantic features $Z^A_{tls}$ and $Z^A_{tun}$ decoupled from TLS traffic and tunnel traffic, respectively. Specifically, DecETT achieves semantic alignment between $Z^A_{tls}$ and $Z^A_{tun}$ by minimizing their cosine similarity loss:

$$\mathcal{L}_{ASA} = -\frac{1}{N}\sum_{i=1}^{N}(1 - \frac{Z^A_{i,tls} \cdot Z^A_{i,tun}}{||Z^A_{i,tls}|| \cdot ||Z^A_{i,tun}||}) \tag{10}$$

By increasing the similarity between $Z^A_{tls}$ and $Z^A_{tun}$ in the high-dimensional semantic space, $Z^A_{tls}$ serves as an additional class-level supervisory signal that provides richer and more stable app-specific information than the class label, thus facilitating more accurate fingerprint generation under encrypted tunnels.

**App Semantic Feature Classification (ASC).** To ensure the correct semantic mapping between the generated fingerprint $Z^A$ and the corresponding app label, we calculate another classification loss as follows:

$$\mathcal{L}_{ASC} = -\frac{1}{N}\sum_{i=1}^{N} y_i(log(\hat{y}^A_{i,tls}) + log(\hat{y}^A_{i,tun})) \tag{11}$$

Therefore, the loss of app semantic feature augmentation submodule $\mathcal{L}_{AFA}$ is calculated as:

$$\mathcal{L}_{AFA} = \lambda_4\mathcal{L}_{ASA} + \lambda_5\mathcal{L}_{ASC} \tag{12}$$

Combined with $\mathcal{L}_{FRD}$, the total loss of DecETT can be summarized as follows:

$$\mathcal{L}_{DecETT} = \mathcal{L}_{FRD} + \mathcal{L}_{AFA} \tag{13}$$

### 4.3 Generated Fingerprint Classification

Once DecETT is well-trained, only $Emb(\cdot)$, $Enc^A_{tun}$ and the tunnel traffic flows $F_{tun}$ instead of parallel flow pairs are needed to generate corresponding fingerprints. This allows DecETT to be employed in real network environments, since parallel flows are inaccessible in real-world deployment. Formally, for a tunnel flow $F_{tun} = \{p_1, p_2, \ldots, p_n\}$, the corresponding fingerprint $FP_{F_{tun}}$ can be generated as:

$$FP_{F_{tun}} = Enc^A_{tun}(Emb(F_{tun})) \tag{14}$$

Ultimately, the corresponding AF result can be calculated as:

$$y_{pred} = Classifier(FP_{F_{tun}}) \tag{15}$$

## 5 Experiments

In this section, we perform empirical evaluations to demonstrate the effectiveness of our proposed framework. We first provide the dataset collection and composition, and then introduce the experimental setup, including baselines, evaluation metrics and implementation details. Finally, we proceed to detail the experimental results and their analysis.

**Table 1: Details of 5 evaluation datasets. TLP refers to the abbreviation of Transport Layer Protocol used by corresponding tunnel protocol.**

| Dataset | TLP | #Apps | #Flows | #Payloads |
|---|---|---|---|---|
| Shadowsocks | TCP | 54 | 346,388 | 29.70G |
| ShadowsocksR | TCP | 54 | 346,418 | 22.78G |
| V2Ray | TCP | 54 | 339,667 | 23.28G |
| Trojan | TCP | 54 | 346,378 | 29.13G |
| OpenVPN | UDP | 54 | 346,296 | 28.14G |

## 5.1 Dataset

DecETT utilizes parallel TLS and tunnel flow pairs to realize accurate app fingerprinting. Although there have been previous related studies, datasets that provide parallel flow pairs have not been established yet. Consequently, we first select 5 representative encrypted tunnels and 54 widely-used apps for our study, and invite several volunteers to interact with these apps through the 5 tunnels separately, thereby producing corresponding traffic flows. In order to purify the collected traffic without noise flows generated by other apps, we follow the traffic collection framework proposed in [15] that uses iptables and NFLOG to mirror and capture pure TLS traffic generated by specific app. The detailed information of 5 datasets is described in Table 1, the configurations of 5 tunnels can be found in Appendix B, and the full list of apps is shown in Appendix C.

## 5.2 Experimental Setup

**Comparison Methods.** We compare our proposed DecETT with four categories of AF or encrypted tunnel traffic analysis methods, including (1) Statistical-based method (AppScanner[30]), which extracts time or packet-related statistical features for further classification; (2) Server information-related method (i.e. FlowPrint[33]) where the communicated server information is considered; (3) Payload-based methods (ET-BERT[17], YaTC[43]) which directly use the raw packet payload content to achieve AF, and (4) Sequence-based methods, such as DF[29], FS-Net[18] and GraphDApp[28], that dedicate to mining the flow sequences for accurate AF.

**Evaluation Metrics.** In this paper, we choose the four widely-used metrics in multi-class classification tasks, i.e., Accuracy, Precision, Recall, and F1-score, to comprehensively evaluate the performance of different methods on AF under encrypted tunnels.

**Implementation Details.** We conduct our evaluation on a server with two Intel(R) Xeon(R) Gold 6240R CPU @2.40 GHz processors, Ubuntu 20.04, 64GB RAM. An NVIDIA Tesla A800 GPU with 80GB VRAM is used to accelerate the computations. Our method is implemented based on Python 3.8.16 and PyTorch 1.12.1+cu113. As for the hyper-parameters, we set the mini-batch size as 256, the hidden size of GRU as 128, the embedding size as 3000, the flow sequence length as 200, and the five loss weights $\lambda_i$ as 1. For all the baselines, we follow their official implementations.

## 5.3 Analysis of AF Results Under Single Tunnel

Firstly, we evaluate the performance of all the comparison methods on accurate app fingerprinting under the specific single tunnel. The corresponding results are reported in Table 2.

### 5.3.1 Main Evaluation.
From Table 2, we can draw the following conclusions:

(1) In terms of the four comprehensive evaluation metrics, our approach DecETT outperforms all the other comparison methods by significant margins. Specifically, DecETT achieves the best performance of 94.2% Accuracy, Recall, and F1-score under ShadowsocksR. The following method is FS-Net, which reaches the F1-score of 61% under V2Ray and around 85% under the other tunnels. The performance of FlowPrint is the worst among all the comparison methods, with nearly all metrics lower than 10% under all five tunnels.

(2) DecETT shows more significant performance superiority under tunnels with more complicated obfuscation. Results of various methods across 5 tunnels show that V2Ray employs more obfuscated encapsulation to the raw TLS traffic. Under the other 4 tunnels, DecETT achieves a performance improvement of approximately 7% to 10% compared to the second best-performed method, FS-Net, while under the V2Ray tunnel, the performance gap rises to nearly 20%. By decoupling the app-irrelevant protocol features and enhancing the fingerprint representations through semantic-shared TLS traffic, DecETT minimizes the negative impact caused by the re-encapsulation mechanism, thereby significantly improving AF performance under complex tunnels.

(3) The performance of both statistical and server information-based methods is not satisfactory enough. Specifically, AppScanner achieves nearly 100% Precision, but fails in Recall value of only around 30% to 60%, indicating its insufficient capability in fully characterizing app-specific information from tunnel traffic. The server information-based method FlowPrint is also ineffective, with an average F1-score of only 1.4% across five tunnels. FlowPrint relies on the flow interaction relationships with various app servers; however, server information is no longer invisible in tunnel traffic, thus resulting in severe performance degradation. These results indicate that statistical and server information-based features cannot provide sufficient flow representation as flow sequences used in DecETT for accurate AF under encrypted tunnels.

(4) As for the two payload-based methods, ET-BERT performs poorly across five datasets, with the highest F1-score of only 25.6%. YaTC achieves better performance than ET-BERT, but still has the maximum performance gap of approximately 40% compared to DecETT. These methods rely on specific plaintext fields in TLS protocol or the pseudo-randomness of encryption algorithms to construct app fingerprints. However, compared to flow sequences, the impact of the re-encapsulation mechanism on these two features is more pronounced and difficult to model, rendering these methods insufficient for effectively modeling app fingerprints under tunnels. These results further highlight the superiority of using flow sequences as the form of tunnel traffic representation in DecETT.

(5) Sequence-based methods perform better than other approaches, with FS-Net achieving the second-best performance across four tunnels. Reasons can be owing to that although the re-encapsulation mechanism affects the packet lengths, changes in flow sequences and packet transmitting directions stay relatively stable compared to the packet payload. Based on utilizing flow sequence as the form of traffic representations, DecETT introduces TLS traffic to provide stronger app-specific information for fingerprint learning, and further decouples the app semantic features hidden in the raw tunnel traffic, thereby achieving accurate app fingerprinting.

**Table 2: Performance comparison results w.r.t. Accuracy (Acc), Precision (P), Recall (R) and F1-score (F1) under 5 tunnels. Bold represents the best and underline refers to the second.**

| Method | Dataset | Shadowsocks | | | | ShadowsocksR | | | | V2Ray | | | | Trojan | | | | OpenVPN | | | |
|---|---|---|---|---|---|---|---|---|---|---|---|---|---|---|---|---|---|---|---|---|---|
| | Metric | Acc | P | R | F1 | Acc | P | R | F1 | Acc | P | R | F1 | Acc | P | R | F1 | Acc | P | R | F1 |
| Statistic | AppScanner[30] | 0.630 | **0.995** | 0.630 | 0.764 | 0.631 | **0.996** | 0.631 | 0.767 | 0.295 | **0.993** | 0.295 | 0.429 | 0.609 | **0.996** | 0.609 | 0.748 | 0.582 | **0.995** | 0.582 | 0.725 |
| Server | FlowPrint[33] | 0.122 | 0.015 | 0.122 | 0.027 | 0.053 | 0.003 | 0.053 | 0.005 | 0.103 | 0.013 | 0.103 | 0.022 | 0.050 | 0.008 | 0.050 | 0.012 | 0.027 | 0.001 | 0.027 | 0.002 |
| Payload | ET-BERT[17] | 0.079 | 0.085 | 0.079 | 0.045 | 0.098 | 0.134 | 0.098 | 0.085 | 0.055 | 0.072 | 0.055 | 0.032 | 0.280 | 0.216 | 0.203 | 0.203 | 0.265 | 0.300 | 0.265 | 0.256 |
| | YaTC[43] | 0.596 | 0.656 | 0.596 | 0.592 | 0.771 | 0.825 | 0.771 | 0.785 | 0.436 | 0.496 | 0.436 | 0.407 | 0.602 | 0.678 | 0.602 | 0.606 | 0.884 | 0.934 | 0.884 | 0.899 |
| Sequence | DF[29] | 0.739 | 0.746 | 0.739 | 0.738 | 0.762 | 0.764 | 0.762 | 0.760 | 0.656 | 0.659 | 0.656 | 0.651 | 0.726 | 0.730 | 0.726 | 0.724 | 0.816 | 0.818 | 0.816 | 0.816 |
| | FS-Net[18] | 0.845 | 0.837 | 0.838 | 0.837 | 0.856 | 0.849 | 0.850 | 0.849 | 0.610 | 0.610 | 0.606 | 0.610 | 0.822 | 0.828 | 0.822 | 0.823 | 0.876 | 0.874 | 0.873 | 0.874 |
| | GraphDApp[28] | 0.786 | 0.800 | 0.786 | 0.789 | 0.817 | 0.812 | 0.811 | 0.812 | 0.503 | 0.516 | 0.501 | 0.516 | 0.767 | 0.763 | 0.760 | 0.763 | 0.810 | 0.805 | 0.806 | 0.805 |
| Ours | DecETT | **0.925** | 0.926 | **0.925** | **0.925** | **0.942** | 0.942 | **0.942** | **0.942** | **0.802** | 0.803 | **0.802** | **0.801** | **0.920** | 0.922 | **0.920** | **0.921** | **0.941** | 0.941 | **0.941** | **0.941** |

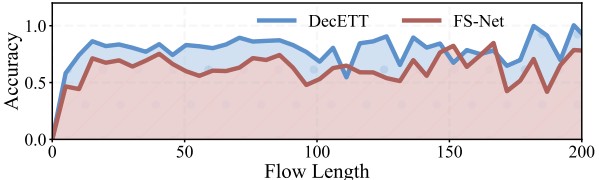

**Figure 5: Comparison results of different flow lengths.**

### 5.3.2 Performance Analysis on Short Flows.
To better illustrate the superiority of DecETT, we analyze its AF performance on flows of varying lengths. Figure 5 shows the Accuracy results of both DecETT and FS-Net on V2Ray flows with lengths ranging from 0 to 200. As shown in Figure 5, DecETT demonstrates a remarkable improvement in fingerprinting short flows with lengths below 100 compared to FS-Net. This improvement can be attributed to both the introduction of TLS traffic as the semantic anchor and the decoupling of tunnel information. By correlating each tunnel flow with its corresponding semantic-shared TLS flow, DecETT provides richer app-specific information than simple label-based approaches. This information is particularly important for short flows, which are easier to suffer from insufficient feature extraction due to their limited lengths. Additionally, the decoupling of tunnel features further mitigates the negative impact caused by tunnel mechanism. Therefore, DecETT can be effectively utilized in AF scenarios that are sensitive to short flows, such as gambling activity detection[12].

### 5.3.3 Visualization.
In addition to the above quantitative evaluations, we conduct a qualitative visualization to further discuss the performance of DecETT. Figure 6 shows the t-SNE[32] visualization of random 5 app fingerprints learned by FS-Net and DecETT under V2Ray, respectively. It can be observed that DecETT enables a more significant aggregation of fingerprints from the same app compared to FS-Net while reducing the overlapping area of fingerprints from different apps, thereby achieving better AF results.

## 5.4 Analysis of AF Results Under Mixed-Tunnel
Evaluation in the previous section is conducted under a specific single tunnel. However, due to the diversity of encrypted tunnels, it is not always feasible to know the exact type of tunnel traffic in advance in real network environments. To address this issue, in this section, we further evaluate the AF performance of DecETT and comparison methods under mixed tunnels.

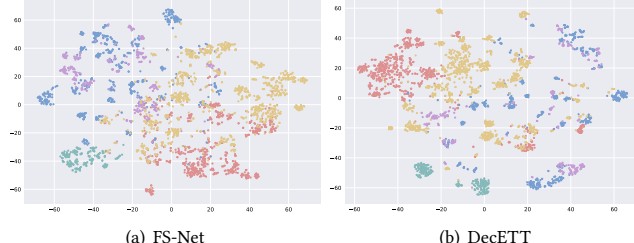

(a) FS-Net     (b) DecETT

**Figure 6: Visual distinction of generated fingerprints where different colors stand for different classes.**

**Table 3: Performance comparison results w.r.t. Accuracy, Precision, Recall and F1-score on mixed-tunnels. Bold represents the best and underline refers to the second.**

| | Method | Accuracy | Precision | Recall | F1-score |
|---|---|---|---|---|---|
| Statistic | AppScanner[30] | 0.542 | **0.996** | 0.542 | 0.694 |
| Server | FlowPrint[33] | 0.075 | 0.015 | 0.075 | 0.023 |
| Payload | ET-Bert[17] | 0.102 | 0.126 | 0.107 | 0.099 |
| | YaTC[43] | 0.601 | 0.652 | 0.601 | 0.603 |
| Sequence | DF[29] | 0.741 | 0.745 | 0.741 | 0.739 |
| | FS-Net[18] | 0.785 | 0.790 | 0.785 | 0.786 |
| | GraphDApp[28] | 0.656 | 0.661 | 0.656 | 0.650 |
| Ours | DecETT | **0.842** | 0.844 | **0.842** | **0.842** |

To conduct this evaluation, we first mix the flows of five encrypted tunnels, where flows generated by the same app share the same label, regardless of whether they are forwarded by the same encrypted tunnel. Each method is required to extract unified app fingerprints from the mixed tunnel traffic. Table 3 concludes the performance of DecETT and other comparison methods. As can be seen from the table, DecETT still outperforms all other baselines under mixed tunnels, achieving 84.2% on the four evaluation metrics. GraphDApp, which relies on burst division, shows significant performance degradation compared to the single-tunnel scenario. This may be due to the fact that different tunnels may employ different packet-sending strategies, thus leading to different burst division results. DF and FS-Net demonstrate relatively better stability, in which FS-Net achieves an F1-score of 78.6%. These results further highlight the superiority of DecETT. By decoupling

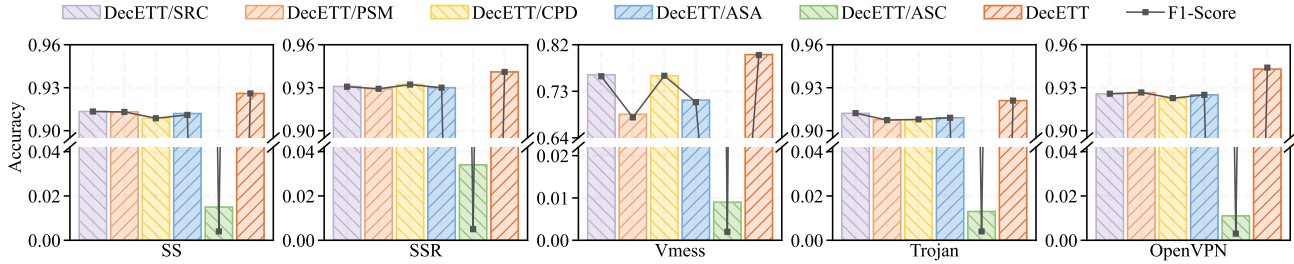

**Figure 7: Ablation study results of key components in DecETT w.r.t. Accuracy and F1-score on 5 tunnel datasets.**

the app-irrelevant tunnel information from flow representations, DecETT enables the model to focus on learning unified app-specific representations across various tunnels, and further provides TLS traffic as robust semantic anchor , thereby achieving more accurate app fingerprinting in real and complex network environment.

## 5.5 Ablation Study

To validate the effectiveness of DecETT, we conduct an ablation study by evaluating its variants, i.e., DecETT/SRC, DecETT/PSM, DecETT/CPD, DecETT/ASA, and DecETT/ASC, to indicate its superiority sufficiently. Figure 7 shows all results of the ablation study.

(1) After removing SRC, the performance of DecETT/SRC declines by 1% to 4% across 5 tunnels, which can be owing to the lack of constraints on decoupling features to fully retain the information in original flow sequences.

(2) Compared to DecETT, both DecETT/PSM and DecETT/CPD show performance decreases, with average F1-score losses of 3.56% and 2.01%, respectively. These results further indicate that decoupling app-irrelevant tunnel features to lower their negative impact on fingerprint generation is essential for accurate AF under tunnels.

(3) The removal of ASA has the most significant impact on DecETT compared with other components despite ASC, with a maximum F1-score drop of 9% under V2Ray. This result demonstrates the importance of stronger app-specific information provided by TLS traffic in accurate AF.

(4) After removing ASC, the performance of DecETT drops drastically, with a maximum F1-score of only 0.5%, highlighting the importance of label supervision in feature decoupling. Without app labels as supervisory signals, DecETT/ASC fails to distinguish useful semantic features for downstream fingerprinting task, resulting in meaningless feature decoupling.

(5) Our model performance get worse by removing any key components, which proves that each of them contributes to the improvement in accurate AF under encrypted tunnels. Furthermore, the performance gaps between DecETT and its variants are further widened when confronting tunnels with more complex obfuscation, such as V2Ray, highlighting its powerful AF capability against tunnel mechanism.

## 5.6 Sensitivity Analysis

In this section, we perform sensitivity analysis on the critical hyperparameter in DecETT, the flow sequence length, which determines the amount of flow sequence information DecETT can utilize for fingerprint learning. In the experimental setup for this section,

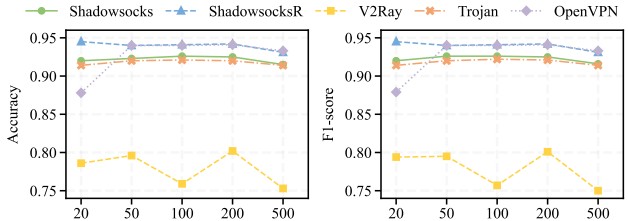

**Figure 8: Sensitivity analysis of DecETT with different flow sequence lengths on 5 tunnel datasets.**

only the flow sequence length is varied, while all other parameters remain the same as previously described.

Figure 8 shows the results under five tunnels. From this figure, we can observe that: (1) DecETT maintains stable performance across different flow sequence lengths, consistently outperforming other comparison methods shown in Table 2; (2) DecETT still achieves remarkable performance even with relatively short flow sequence length (e.g., length=20), highlighting its strong capability in accurate fingerprint construction; (3) Excessively long flow sequences lead to performance decline. This can be attributed to that the later stages of flow transmission mainly focus on transmitting large amounts of data, resulting in packet length sequences with high similarity (e.g., numerous packets of MTU size). Overall, we thus conclude that DecETT is relatively insensitive to different flow sequence lengths, demonstrating its robustness to hyperparameter perturbations.

## 6 Conclusion

In this work, we propose DecETT, a dual decouple-based semantic enhancement method to achieve accurate app fingerprinting under encrypted tunnels. Considering the negative impact caused by re-encapsulation mechanism of encrypted tunnels on accurate fingerprint extraction, we first introduce TLS traffic as a relatively stronger and robust semantic anchor to enhance fingerprint learning, and further decouple the protocol features and app semantic features to reduce the impact of encrypted tunnels in fingerprint generation. Finally, the decoupled app semantic features are utilized for fingerprints generation and classification. Experiments under five representative encrypted tunnels indicate that DecETT outperforms state-of-the-art methods in accurate AF under encrypted tunnels by significant margins, and further demonstrates its superiority under tunnels with more complicated obfuscation.

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

## A    RE-ENCAPSULATION MECHANISM ILLUSTRATION OF ENCRYPTED TUNNELS

In section 3.2, we analyze the source code of the re-encapsulation mechanism summarized from Shadowsocks to illustrate the independence of tunnel features and app semantic features. In this section, we extend the discussion to the other four tunnels: ShadowsocksR, V2Ray, Trojan, and OpenVPN. Some of these tunnels are implemented by different programming languages, such as Go, C and C++, which are not as concise as Python used by Shadowsocks. As a result, the source code pipeline can be too lengthy to be fully presented in this paper. To this end, we provide a brief overview of the other four re-encapsulation mechanisms together with the corresponding source code link for interested readers.

In summary, the other four tunnels also forward data by maintaining two socket communications and their correlation. The core difference lies in the encryption algorithms and protocols used for re-encapsulation: (1) ShadowsocksR[1] employs the same re-encryption mechanism as Shadowsocks. (2) V2Ray[2], on the other hand, uses closures *requestDone()* and *responseDone()* operations to implements the re-encapsulation mechanism, in which the encryption algorithms and encapsulation details follows its private protocol Vmess. (3) Trojan[3] conceals its traffic characteristics using the standard SSL protocol, and applies the SSL mechanism in *Boost.Asio* for re-encapsulation of the forwarded data. (4) OpenVPN[4] also implements its re-encapsulation mechanism based on OpenSSL protocol, while further developing its private protocol, OpenVPN, on top of OpenSSL.

Overall, the varied re-encapsulation mechanisms pose challenges to accurate app fingerprinting under encrypted tunnels. However, their reliance on socket communication underscores the generalizability and correctness of decouple-based AF methods across various encrypted tunnels.

## B    CONFIGURATIONS OF FIVE ENCRYPTED TUNNELS

Table 4 provides detailed configurations of the five encrypted tunnels used in our experiments. In the following, we illustrate each of the configuration in detail.

- **Encrypted Algorithm(EA).** Encrypted algorithm refers to the algorithm during the re-encryption of the forwarded traffic data. In our experiments, ShadowsocksR uses AES-256-CFB as the encrypted algorithm, while the other four tunnels use AES-256-GCM.
- **Protocol.** Protocol refers to the specific communication protocol used by the encrypted tunnel, which determines the way of data re-encapsulation and transmission between tunnel client and tunnel server. Some encrypted tunnels use their specific private protocols, such as Origin used by ShadowsocksR, Vmess used by V2Ray, and OpenVPN protocol used by OpenVPN.

- **Obfuscation(Obfs).** Obfuscation refers to techniques used to disguise the existence of the encrypted tunnel by modifying the appearance of the traffic, making it harder to detect. Obfuscation can be achieved by altering packet characteristics or mimicking other types of traffic.
- **Notes.** OpenVPN provide two different tunneling modes, TUN mode and TAP mode. TUN mode operates at the network layer and is designed for routing IP packets, while TAP mode operates at the data link layer, which emulates a virtual Ethernet adapter. Since we focus on the application fingerprinting, we choose TUN mode, which is more suitable for this scenario, to implement OpenVPN.

**Table 4: Detailed configurations of 5 encrypted tunnels in our evaluation.**

| Tunnel | EA | Protocol | Obfs | Notes |
|---|---|---|---|---|
| Shadowsocks | AES-256-GCM | SOCKS | - | - |
| ShadowsocksR | AES-256-CFB | Origin | tls1.2_ticket_auth | - |
| V2Ray | AES-128-GCM | Vmess | - | - |
| Trojan | AES-128-GCM | HTTPS | - | - |
| OpenVPN | AES-128-GCM | OpenVPN | - | TUN Mode |

## C    FULL LIST OF MOBILE APPS

We provide a full list of 54 mobile apps selected in our experiments (see Table 5).

**Table 5: Full list of the mobile apps.**

| No. | Package Name | No. | Package Name |
|---|---|---|---|
| 1 | air.tv.douyu.android | 28 | com.snapchat.android |
| 2 | cn.xdf.woxue.student | 29 | com.sohu.sohuvideo |
| 3 | com.amazon.mShop.android.shopping | 30 | com.ss.android.article.video |
| 4 | com.bilibili.app.in | 31 | com.ss.android.ugc.aweme |
| 5 | com.bilibili.comic | 32 | com.ss.android.ugc.trill |
| 6 | com.bittorrent.client | 33 | com.talk51.international |
| 7 | com.duowan.kiwi | 34 | com.taobao.idlefish |
| 8 | com.duowan.mobile | 35 | com.taobao.live |
| 9 | com.facebook.katana | 36 | com.taobao.taobao |
| 10 | com.google.android.youtube | 37 | com.tencent.androidqqmail |
| 11 | com.huajiao | 38 | com.tencent.mm |
| 12 | com.hunantv.imgo.activity | 39 | com.tencent.mobileqq |
| 13 | com.larksuite.suite | 40 | com.tencent.qqlive |
| 14 | com.meelive.ingkee | 41 | com.tencent.qqmusic |
| 15 | com.mogujie | 42 | com.tencent.weread |
| 16 | com.netease.cc | 43 | com.tmall.wireless |
| 17 | com.netease.edu.study | 44 | com.vipkid.ark.international.parent |
| 18 | com.nhn.android.nmap | 45 | com.xes.jazhanghui.activity |
| 19 | com.periscope.pscp | 46 | com.xiaomi.shop |
| 20 | com.pplive.androidphone | 47 | com.xingin.xhs |
| 21 | com.qihoo360.mobilesafe | 48 | com.xunlei.downloadprovider |
| 22 | com.qiyi.video | 49 | com.xunmeng.pinduoduo |
| 23 | com.sdu.didi.psnger | 50 | com.yandex.browser |
| 24 | com.shanbay.sentence | 51 | com.youku.phone |
| 25 | com.sina.weibo | 52 | com.zhihu.android |
| 26 | com.skype.raider | 53 | me.ele |
| 27 | com.smile.gifmaker | 54 | ru.ok.android |

---

[1]https://github.com/shadowsocksrr/shadowsocksr/(shadowsocks/tcprelay.py)
[2]https://github.com/v2fly/v2ray-core/(proxy/vmess/outbound/oubound.go)
[3]https://github.com/trojan-gfw/trojan(/src/session/clientsession.cpp)
[4]https://github.com/OpenVPN/openvpn/(src/openvpn/ssl_openssl.c)

