# OpenReview forum: "DecETT: Accurate App Fingerprinting Under Encrypted Tunnels via Dual Decouple-based Semantic Enhancement"
_ACM.org/TheWebConf/2025/Conference — WWW 2025 Poster_

### Official Review · Reviewer_atiG · 2024-11-20

**Novelty:** 5
**Technical Quality:** 5

**Review:**

This paper presents DecETT, a dual decouple-based semantic enhancement method aimed at achieving accurate application fingerprinting (AF) under encrypted tunnels. The authors introduce a novel approach that utilizes TLS traffic as a semantic anchor and decouples protocol features from app-specific features to mitigate the negative impacts of tunnel obfuscation. The paper includes extensive experiments across five widely-used encrypted tunnels, demonstrating the effectiveness of DecETT compared to existing methods.

### **Pros**
1. The introduction of a dual decouple-based method is a good contribution to the field of application fingerprinting, addressing the challenges posed by encrypted tunnels.
2. The authors conducted comprehensive experiments across multiple encrypted tunnel types, providing a solid empirical foundation for their claims. The results indicate DecETT’s effectiveness.

------

### **Cons**
1. The model relies on TLS traffic as a semantic anchor, but in some scenarios, the original TLS traffic may not be available, which may limit its applicability.
2. The dataset primarily consists of traffic generated by volunteers, which may not represent the diversity and noise of real-world network traffic. Moreover, the assumption that only one application is active per session oversimplifies real-world multi-tasking environments.
3. DecETT may lack robustness against adaptive and adversarial example attacks, where deliberate traffic manipulation or advanced obfuscation techniques could compromise its feature extraction and classification accuracy. This needs to be further discussed.

**Questions:**

1. Can the model handle noisy, mixed-application traffic in real-world environments where multiple applications are active simultaneously?
2. How would DecETT perform against adaptive or adversarial example attacks, and what mechanisms could be incorporated to enhance its robustness in such scenarios?

**Reviewer Confidence:**

3: The reviewer is confident but not certain that the evaluation is correct

**Scope:**

3: The work is somewhat relevant to the Web and to the track, and is of narrow interest to a sub-community

---

### Official Review · Reviewer_PAPx · 2024-11-22

**Novelty:** 4
**Technical Quality:** 4

**Review:**

The paper introduces DecETT, a novel approach to app fingerprinting (AF) under encrypted tunnels.
Due to the increasing use of encrypted tunnels, existing AF methods struggle due to obfuscation and re-encapsulation mechanisms. DecETT addresses these issues using a dual decouple-based method to separate app-specific and tunnel-specific features, utilizing TLS traffic as a semantic anchor to enhance fingerprint generation. Evaluation results under five encrypted tunnels demonstrate that DecETT outperforms state-of-the-art methods in accuracy, particularly in scenarios involving complex obfuscation.

I’m not an expert in this field. I feel that the dual decouple-based approach presented in the paper to be innovative, and the experiments are thorough. However, I do have a few concerns.

My primary concern lies in the applicability of the method in real-world scenarios. First, tusing TLS traffic as a semantic anchor may not always be feasible in scenarios where this traffic is unavailable or obfuscated further. In practice, TLS traffic may be unavailable due to various reasons (e.g., network policy restrictions), which limits the applicability of DecETT in some environments. Second, the design of DecETT appears to be quite complex, employing a dual-branch Siamese network along with multiple loss functions to achieve feature decoupling and enhancement, which may result in significant computational and time costs. The issue of the costs, and how to efficiently implement this method on resource-constrained network devices or edge computing nodes is not sufficiently discussed.

The paper mentions that different tunnels may use different encapsulation and forwarding strategies, but it does not sufficiently validate whether DecETT can consistently perform well when these strategies change. For instance, there is insufficient experimental evidence regarding the robustness of the method against different types of traffic feature obfuscation, such as time-based strategies.

**Questions:**

1.Can the approach be deployed in real-world environments?

  2.How robust is the approach?

**Reviewer Confidence:**

2: The reviewer is willing to defend the evaluation, but it is likely that the reviewer did not understand parts of the paper

**Scope:**

3: The work is somewhat relevant to the Web and to the track, and is of narrow interest to a sub-community

---

### Official Review · Reviewer_zuo7 · 2024-11-28

**Novelty:** 3
**Technical Quality:** 5

**Review:**

## Paper Summary

This paper proposes DecETT, a dual decouple-based semantic enhancement method aimed at improving the accuracy of app fingerprinting under encrypted tunnels. First, it introduces traffic correlation and embedding to guide and enhance fingerprint generation. Second, it decouples the flow representation by separately processing tunnel-related features and app semantic features, thereby lowering the negative impact of the re-encapsulation mechanism on fingerprint extraction and subsequent classification.

The experiments conducted under five widely used encrypted tunnels demonstrate that DecETT outperforms state-of-the-art solutions. The contributions of this work lie in proposing an effective AF method that achieves higher accuracy in encrypted traffic analysis.

## Strengths

- The paper identifies the negative impact of re-encapsulation mechanisms on fingerprint generation and app fingerprint (AF) classification. It summarizes a clear threat model and proposes decoupling app fingerprints from tunnel sequences to mitigate this impact and enhance accuracy.
- The proposed methodology involves a novel embedding approach to preprocessing, followed by flow representation decoupling and app semantic feature augmentation. These are well-documented, with no apparent flaws in the description. The evaluation results support its validity.
- The system has been evaluated on a comprehensive dataset, demonstrating significant improvements over competing solutions in both single tunnel and mixed tunnel scenarios.

## Weaknesses

- The related work section lacks a clear distinction between the proposed system and other existing systems.
- Experimental setups may introduce bias, raising concerns about the system’s generalization and robustness.
- The mathematical formulas used in the methodology are not rigorously proven or explained.
- The system’s contributions are incremental, with limited novelty.
- The system does not support composite fingerprints, limiting its applicability.

## Detailed Comments

### Methodology

The methodology appears largely correct, and the experimental data supports this. However, the formulas introduced in the paper should be further validated or explained in the appendix to enhance the paper’s quality. While the approach seems sound, the novelty or challenge of the system is unclear. Additional clarifications are expected.

### Evaluation

The experimental environment leverages IP tables to remove noise from other applications. However, the system’s robustness in noisy environments is neither tested nor addressed.

The selected packages in the dataset lack detailed descriptions, such as their versions, complexity, types, primary operators, and target audiences. It is also unclear how the authors made the selection and why they selected 54 apps. The omission makes it difficult to assess the representativeness of the dataset. A large proportion of the applications are from major Chinese Internet companies (e.g., Tencent, NetEase, Alibaba, etc.). This limits the system’s generalization to other contexts.

Ablation studies should be supplemented with an additional experiment showing results in mixed tunnel scenarios.

### Presentation

The paper is generally well-written, with clear logic and structure. Below are a few minor issues identified during the review process.

- Figure 2 should be replaced with a proper listing instead of a screenshot of code. Additionally, the figure is smaller in size and not centered compared to other diagrams.
- Line 206: There is no space after the period.
- Line 380: Would it be S_{Port} instead of D_{port}?
- Line 669: server information is "no longer invisible" ...

**Questions:**

- The authors mentioned that there is no publicly available dataset and provided limited results in the artifact. Are the authors willing to release the source code and dataset for future research?
- How were those 54 apps selected for evaluation?
- The technical approach and artifact provided seem simplistic, making it difficult to identify the novelty or challenge of this work. Could the authors elaborate further?
- The paper claims that DecETT is relatively insensitive to different flow sequence lengths. Why is V2Ray significantly more affected? Could the authors provide additional experiments with flow sequence lengths below 20?

**Reviewer Confidence:**

3: The reviewer is confident but not certain that the evaluation is correct

**Scope:**

3: The work is somewhat relevant to the Web and to the track, and is of narrow interest to a sub-community